# Solid Lipid Nanoparticles Based on Monosubstituted Pillar[5]arenes: Chemoselective Synthesis of Macrocycles and Their Supramolecular Self-Assembly

**DOI:** 10.3390/nano12234266

**Published:** 2022-11-30

**Authors:** Darya Filimonova, Anastasia Nazarova, Luidmila Yakimova, Ivan Stoikov

**Affiliations:** 1A.M. Butlerov Chemistry Institute, Kazan Federal University, 18 Kremlyovskaya Str., 420008 Kazan, Russia; 2Federal State Budgetary Scientific Institution «Federal Center for Toxicological, Radiation, and Biological Safety», Nauchny Gorodok-2, 420075 Kazan, Russia

**Keywords:** macrocycles, pillar[5]arene, self-assembly, solid lipid nanoparticles

## Abstract

Novel monosubstituted pillar[5]arenes with one or two terminal carboxyl groups were synthesized by the reaction of succinic anhydride with pillar[5]arene derivative containing a diethylenetriamine function. The ability for non-covalent self-assembly in chloroform, dimethyl sulfoxide, as well as in tetrahydrofuran-water system was studied. The ability of the synthesized macrocycles to form different types of associates depending on the substituent nature was established. The formation of stable particles with average diameter of 192 nm in chloroform and of 439 nm in DMSO was shown for pillar[5]arene containing two carboxyl fragments. Solid lipid nanoparticles (SLN) based on monosubstituted pillar[5]arenes were synthesized by nanoprecipitation in THF-water system. Minor changes in the structure of the macrocycle substituent can dramatically influence the stability and shape of SLN (spherical and rod-like structures) accordingly to DLS and TEM. The presence of two carboxyl groups in the macrocycle substituent leads to the formation of stable spherical SLN with an average hydrodynamic diameter of 364–454 nm. Rod-like structures are formed by pillar[5]arene containing one carboxyl fragment, which diameter is about of 50–80 nm and length of 700–1000 nm. The synthesized stable SLN open up great prospects for their use as drug storage systems.

## 1. Introduction

Solid lipid nanoparticles (SLN) are an alternative generation of nanoparticles compared with well-known colloidal systems such as liposomes and polymeric micro- and nanoparticles [1,2,3,4]. They are applied in medical and pharmaceutical chemistry. SLN increase absorption and bioactivity, improve tissue distribution in the target organ and ensure controlled drug release [5,6,7,8]. Low toxicity, the capability to encapsulate hydrophilic and hydrophobic substances and the possibility of large-scale production are advantages of these systems. The stated above makes it possible to use SLN for the development of various pharmaceutical products [9,10,11]. Nowadays, the use of macrocyclic compounds as a lipid matrix is attracting more and more attention due to the possibility of molecules double encapsulation either in the host cavity or in the SLN matrix [12,13]. The ability to selective interaction with target cells through host-guest complexation will significantly improve targeted drug delivery. A number of works have proposed amphiphilic molecules, e.g., calix[n]arenes and cyclodextrins, as lipid analogues [14,15,16,17,18]. The use of a promising class of macrocycles, pillar[5]arenes, in the synthesis of SLN was shown for the first time in our research group [19,20,21].

The choice of this class of macrocyclic compounds was based on synthetic availability and possibility of regoiselective functionalization [22,23,24,25]. Moreover, the macrocyclic cavity of pillar[5]arene can take an active part in the formation of host-guest complexes, and also makes it possible to obtain different types of associates (supramolecular polymers, micelles, vesicles, rotaxanes and pseudorotaxanes) [26,27,28,29,30,31,32,33].

The presented work is ongoing studies of the monosubstituted pillar[5]arene as a building block for nanoparticles [19,20,21]. We assumed that minor changes in the substituent structure of macrocycle can dramatically change the stability and shape of the SLN. We chose pillar[5]arenes containing one or two fragments of succinic acid to confirm this hypothesis. This research is presented an approach to the chemoselective synthesis of new monosubstituted pillar[5]arenes containing one or two carboxyl groups. The substituent effect on the aggregation ability of the macrocycle has also been evaluated. Solid lipid nanoparticles based on the obtained compounds were synthesized and the features of supramolecular self-assembly on the SLN size were studied.

## 2. Results and Discussion

### 2.1. Synthesis of Monosubstituted Pillar[5]arenes Containing Amide and Carboxyl Groups

Previously, our research group optimized acylation conditions with succinic anhydride of monosubstituted pillar[5]arenes containing *N*-propylamide and *N*-(aminobutyl)amide fragments [21]. Pillar[5]arene **1** containing one diethylenetriamine fragment, which has primary and secondary amino groups, was chosen as a starting amine for interaction with succinic anhydride. The compound **1** was obtained according to the literature [19] (Figure 1).

It is known that macrocycle **1** tends to form supramolecular polymers in proton-donor chloroform, which is not observed in proton-acceptor solvents such as dimethyl sulfoxide and acetonitrile [19]. Therefore, the reaction was carried out in a tetrahydrofuran-acetonitrile mixture to knock out the *N*-(aminoalkyl)amide substituent from the macrocyclic cavity. The use of acetonitrile led to acylation at both the primary and secondary amino groups in the case of macrocycle **1** containing diethylenetriamine fragment. It is resulting in the formation of monosubstituted pillar[5]arene **2** with two carboxyl groups (Figure 1). The effect of acetonitrile leads to the knock out of the substituent from the cavity, as expected, whereby the secondary amino group becomes sterically accessible and reacts with the anhydride molecule. Monosubstituted pillar[5]arene **2** containing two fragments of succinic acid was obtained in 76% yield. The ^1^H NMR spectrum is in good agreement with the structure of the synthesized macrocycle **2** (Figure 1).

In the ^1^H NMR spectrum of the compound **2** (Figure 1) the proton signals of aromatic fragments and methylene bridges as well as methoxy groups resonate as multiplets in the 6.70–6.83 ppm and 3.47–3.62 ppm, respectively. The H^4^ proton signals of oxymethylene fragments split into two singlets (4.30 and 4.34 ppm). Similar splitting into two triplets is also characteristic for the protons of each amide group. Thus, the H^5^ proton signals resonate as two triplets (8.19 and 8.32 ppm) with *^3^J_HH_* 5.6 and 5.8 Hz. The H^7^ proton signals of amide group also appear as two triplets with chemical shifts of 7.92 and 8.06 ppm and *^3^J_HH_* 5.6 and 5.4 Hz, respectively. Such a signal-doubling is due to the coexistence of E/Z isomers, which appear by the tautomerism of the amide group. Therefore, the amide group either forms a hydrogen bond with the oxygen atom of the carbonyl fragment or it is in the shielding zone of the carbonyl group. This leads to a doubling of the proton signals for each amide fragment. The H^8^ and H^9^ proton signals of the methylene fragments resonate as a multiplet in the 2.26–2.56 ppm range. The H^6^ methylene proton signals of diethylenetriamine fragment overlap with the residual proton signals of water and are also observed as a multiplet at 3.14–3.29 ppm.

It was decided to study the interaction of amine **1** with succinic anhydride in the presence of ammonium chloride to obtain monosubstituted pillar[5]arene containing only one fragment of succinic acid. We suppose that the addition of ammonium chloride to the reaction mixture leads to protonation of the secondary amino group. In view of this the reaction will proceed chemoselectively, and only the primary amino group will undergo acylation. Indeed, the addition of ammonium chloride to the reaction mixture made it possible to direct the reaction with macrocycle **1** only at the primary amino group. The compound **3** was obtained in 74% yield. Structure of all the synthesized compounds was confirmed by a number of physical methods (^1^H NMR, ^13^C NMR and IR spectroscopy). The composition was confirmed by mass spectrometry and elemental analysis (Appendix A). It is worth noting that the melting point for macrocycle **2** is 104 °C and that is 120 °C for pillar[5]arene **3**. Apparently, the lower melting temperature for the compound **2** is caused by the lower packing density of molecules in space compared to the macrocycle **3**. This fact confirms that in the case of pillararene **3** the formation of supramolecular polymers is possible, which leads to denser packing.

### 2.2. Aggregation Properties of Monosubstituted Pillar[5]arenes Containing Carboxyl Groups

One of the important features of monosubstituted pillar[5]arenes is supramolecular self-assembly, which leads to the formation of various oligomeric and polymeric structures [34,35,36,37,38,39]. Therefore, the next stage of the work was studying the aggregation ability of **2** and **3**. It is known that monosubstituted pillar[5]arenes can form different typeCs of supramolecular architectures depending on the solvent used [40,41,42]. The macrocycle **2** contains a bulkier substituent including two fragments of succinic acid. In this regard, we hypothesized that formation of supramolecular polymers by pillar[5]arene **2** is impossible or unprofitable for steric reasons. We studied association properties (Appendix A) of the obtained compounds **2** and **3** in chloroform and DMSO in the 1 × 10^−5^ – 1 × 10^−3^ M concentration range by dynamic light scattering (DLS) to confirm this hypothesis. Pillar[5]arene **2** at C = 1 × 10^−3^ M forms particles (Table 1) with hydrodynamic diameter (d) d_1_ = 289 ± 9 nm in DMSO. The polydispersity index (PDI) of this system is 0.44 ± 0.03. Macrocycle **3** forms systems similar in size and polydispersity index (d_1_ = 332 ± 7 nm, PDI = 0.39 ± 0.03) in the same conditions. A small fraction of larger particles with sizes d_2_ = 5050 ± 254 nm and d_2_ = 5150 ± 128 nm for pillar[5]arenes **2** and **3** respectively, can be additionally observed in both cases (Table 1). Decreasing the concentration of the pillar[5]arene **2** and **3** solutions in DMSO to C = 1 × 10^−4^ M leads to different effect on their self-assembly. A monodisperse system with PDI = 0.26 ± 0.03 and a particle diameter of 439 ± 59 nm was formed (Table 1) in the case of macrocycle **2**. There are no significant changes in the particle sizes and polydispersity index in the case of the compound **3** (Table 1). 

The study of the aggregation properties of the obtained pillar[5]arenes **2** and **3** in chloroform (Appendix A) showed that almost in the entire concentration range (from 1 × 10^−5^ to 1 × 10^−3^ M) macrocycles **2** and **3** form polydisperse systems (Table 1). The exception is the solution of the compound **2** at a 1 × 10^−3^ M concentration (d = 192 ± 20 nm, PDI = 0.26 ± 0.01). Apparently, the presence of two carboxyl functions in the structure of **2** leads to the fact that the pillar[5]arene **2** becomes similar to surfactant molecules in its aggregation properties. The macrocycle can be clearly distinguished between lipophilic (macrocyclic rim) and hydrophilic parts (substituent with two fragments of succinic acid). The aggregation of macrocycle **2** becomes possible only when a certain concentration is reached (1 × 10^−3^ M in CHCl_3_ and 1 × 10^−4^ M in DMSO) due to the similar behavior to surfactants. The formation of polydisperse systems by the compound **3** in chloroform is probably caused by the formation of supramolecular polymers. The solution of pillararene **3** was studied by ^1^H NMR spectroscopy (Figure 2) in two solvents (CDCl_3_ and DMSO-*d_6_*). It is worth noting that the chemical shifts of methylene proton signals of the succinic acid fragment (H^8^ and H^9^) practically do not change regardless of the solvent, whereas, chemical H^5^ and H^6^ proton signals of diethylenetriamine fragment undergo upfield shift for the solution of compound **3** in chloroform. It is due to their strong shielding by macrocyclic cavity. This indicates inclusion of the substituent in the macrocyclic cavity with the formation of supramolecular polymer (Figure 2b).

Figure 3 shows the proposed self-assembly scheme of the macrocycles **2** and **3**.

The next stage of the work was a study of the noncovalent self-assembly patterns of pillar[5]arenes **2** and **3** with the formation of solid lipid nanoparticles (SLN-2 and SLN-3 respectively) in water (Table 2).

Today, a number of works are devoted to the synthesis of different shaped SLN [20,43,44,45,46]. It has been shown that the shape of SLN can change depending on the nature and size of the substituents in the molecule [20,46], the interaction of the solute with the solvent [43], and the loading of SLN with some substance [45]. In this work, we studied the influence of substituent nature in the structure of monosubstituted pillar[5]arenes on the size and stability of formed particles. We obtained SLN in water by nanoprecipitation using the THF-water solvent system according to the literature method [47]. The synthesized nanoparticles were characterized by DLS and electrophoretic light scattering methods (Appendix A). The initial concentration of pillar[5]arenes **2** and **3** was 3 × 10^−4^ M. The most stable SLN are formed at 3 × 10^−4^ M concentration (Table 2) for the macrocycle **2**, as indicated by the zeta potential value (ζ = −33 mV). There is insignificant increase in the polydispersity index of the systems when the solution of the compound **2** is diluted, while the size of associates decreases. A different situation was observed for SLN formed by the pillar[5]arene **3** (Table 2). Both particle enlargement and an increase in the polydispersity index of the systems occur with the concentration decrease of the solutions. At the same time, the smallest aggregates size and PDI value (d = 209 ± 2 nm, PDI = 0.14 ± 0.01) are also characteristic for the highest concentration of 3 × 10^−4^ M as in the case of macrocycle **2**. Pillar[5]arene **2** containing a bulkier substituent, namely, two fragments of succinic acid, tends to form particles of larger size (d = 454 ± 19 nm) compared to the **3** (d = 209 ± 2 nm). The presence of two carboxyl groups in the macrocycle substituent leads to the formation of stable SLN over the studied concentration range within three orders of magnitude from 10^−6^ to 10^−4^ M, which is obviously due to the framing of particles by carboxylate groups. At the same time, the transition from the amido acid residue in the case of the compound **2** to the aminoamido acid fragment in the pillar[5]arene **3** leads to the formation of denser SLN by supramolecular self-assembly of the macrocycle into polymeric structures. It is in good agreement with its melting point and the size of the formed SLN. Measurement of ζ-potential allows to predict the stability of colloidal dispersions during storage. The use of the macrocycle **2** is preferable for obtaining more stable SLN because aggregation of charged particles with a high ζ-potential occurs to a lesser degree due to electrostatic repulsion. A similar pattern of ζ-potential changes has been described for non-macrocyclic surfactants [48].

The synthesized solid lipid nanoparticles were additionally studied by transmission electron microscopy (TEM). The formation of spherical nanosized aggregates with an average diameter of 200 nm was established for the SLN-2 (Figure 4a). Macrocycle **3** formed rod-like particles which diameter is about of 50–80 nm and length of 700–1000 nm (Figure 4b). The data obtained by TEM are in good agreement with the previously advanced hypothesis about the substituent effect on the self-assembly of the synthesized compounds **2** and **3**. The formulation of micelle-like structures is typical for pillar[5]arene **2** containing two carboxyl groups. Supramolecular polymers are formed in the case of macrocycle **3** containing one carboxyl function.

## 3. Conclusions

Monosubstituted pillar[5]arenes containing one or two carboxyl groups were synthesized for the first time with good yields. The chemoselectivity of the process was controlled by adding ammonium chloride. The ability of the synthesized macrocycles to form different types of associates depending on the substituent nature was established. The formation of stable particles with average diameter of 192 nm in chloroform and of 439 nm in DMSO was shown for pillar[5]arene containing two carboxyl fragments. Monosubstituted pillar[5]arene containing one carboxyl function is prone to the supramolecular polymer formation in CDCl_3_. Solid lipid nanoparticles (SLN) based on obtained macrocycles were synthesized and characterized by DLS and TEM. Minor changes in the structure of the macrocycle substituent can dramatically change the stability and shape of SLN (spherical and rod-like structures). The presence of two carboxyl groups in the macrocycle substituent leads to the formation of stable spherical SLN (364–454 nm) while rod-like structures are formed by pillar[5]arene containing one carboxyl fragment. The use of pillar[5]arene with two carboxyl functions is preferable to obtain more stable SLN. Synthesized stable SLN open up great prospects for their use as drug storage systems. This research could be regarded as a starting point for further investigation of the applicability of these materials.

## Data Availability

The data presented in this study are available in Appendix A.

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
