# Peer review of "Solid Lipid Nanoparticles Based on Monosubstituted Pillar[5]arenes: Chemoselective Synthesis of Macrocycles and Their Supramolecular Self-Assembly"

_nanomaterials, 2022, doi:10.3390/nano12234266_

Round 1
Reviewer 1 Report
In this paper, the authors reported the preparation of solid lipid nanoparticles based on the self-assembly of monosubstituted pillar[5]arenes. They found the morphology of the particles could be controlled by different preparation method. They obtained spherical and rod-like structures by nanoprecipitation. Most of the experiments were well conducted. The following are some detailed comments.
1. In Figure 2, the analysis of the NMR spectra may be wrong. Why does the peak of H7 appear between -0.4- -0.8 ppm?
2. In Figure 2, the authors should clearly explain that why there is no peak between 2.0-0 ppm with DMSO-d6?
3. The preparation condition should be added in the Figure 4 caption.
4. The author are suggested to give a more detailed study on the self-assembly behavior. The morphology of the nanoparticles prepared at different condition may be supplied.
Author Response
First of all, we would like to thank respected Reviewer for careful consideration of the manuscript. In accordance with the comments, the following changes have been made:
- In Figure 2, the analysis of the NMR spectra may be wrong. Why does the peak of H7 appear between -0.4- -0.8 ppm?
It was an unfortunate typo. The NMR spectra were rechecked and figure 2 has been modified. The formation of inclusion complexes in paracyclophanes is unambiguously detected by the upfield shift of proton signals in 1H NMR spectra. This makes it possible to determine both the formation of host–guest complexes and to measure low values of association constants by 1H NMR spectroscopy [https://doi.org/10.1039/D0PY00327A; https://doi.org/10.1039/C4CC01403K; https://doi.org/10.1039/D1CC03010H]. The formation of supramolecular polymers is well known for monosubstituted pillar[5]arenes depending on the solvent used [https://doi.org/10.1039/C8CC08085B; https://doi.org/10.1016/j.trechm.2020.07.004]. Some publications show the use of DMSO for preventing the formation of supramolecular polymers [https://doi.org/10.1021/cr500633b; https://doi.org/10.1021/acs.jpcb.5b02515]. It is clearly seen in the 1H NMR spectra, where substituent proton signals resonate as sharp, not broadened signals in the downfield. On the contrary, the formation of supramolecular polymers was detected in case of chloroform as a solvent, due to the inclusion of a macrocycle substituent in the cavity of the neighboring pillar[5]arene. Shielding of the substituent by the electron-donor cavity of neighboring macrocycle leads to upfield shift of the substituents proton signals.
- In Figure 2, the authors should clearly explain that why there is no peak between 2.0-0 ppm with DMSO-d6?
The formation of supramolecular polymers is well known for monosubstituted pillar[5]arenes depending on the solvent used [https://doi.org/10.1039/C8CC08085B; https://doi.org/10.1016/j.trechm.2020.07.004]. Some publications show the use of DMSO for preventing the formation of supramolecular polymers [https://doi.org/10.1021/cr500633b; https://doi.org/10.1021/acs.jpcb.5b02515]. It is clearly seen in the 1H NMR spectra, where substituent proton signals resonate as sharp, not broadened signals in the downfield. The following sentence was added to the text of manuscript: “Whereas, a chemical H5 and H6 proton signals of diethylenetriamine fragment undergo upfield shift for the solution of compound 3 in chloroform. It is due to their strong shielding by macrocyclic cavity. This indicates inclusion of the substituent in the macrocyclic cavity with the formation of supramolecular polymer (Fig. 2b)”. In addition, figure 2 has been modified.
- The preparation condition should be added in the Figure 4 caption.
Sample preparation conditions were added to the caption for figure 4.
- The author are suggested to give a more detailed study on the self-assembly behavior. The morphology of the nanoparticles prepared at different condition may be supplied.
Self-assembly of synthesized solid lipid nanoparticles formed by macrocycles 2 and 3 was studied by DLS (Table 2) and TEM. TEM images made it possible to establish the effect of the pillar[5]arene structure on the shape of formed particles. The particles morphology is preserved regardless of concentration.
TEM images of: a) SLN-2 (3 × 10-6 M); b) SLN-2 (3 × 10-5 M); c) SLN-3 (3 × 10-6 M); d) SLN-3 (3 × 10-5 M) prepared by nanoprecipitation using the THF-water solvent system.

Reviewer 2 Report
In this manuscript, the authors reported the selective synthesis of two different monosubstituted pillar[5]arenes with one or two carboxyl groups by optimizing the reaction conditions (with or without an ammonium salt). In addition, they revealed the formation of several aggregates in organic solvents or aqueous solution by NMR, DLS and TEM analyses. The microscopic images obtained by TEM analysis were clear and well consistent with proposed aggregate structures shown in Figure 3. Also, the formation of supramolecular polymers was clearly suggested by 1H NMR spectrum shown in Figure 2. I think these results are interesting and of interest to readers. Therefore, I believe that this manuscript can be published in this journal after revision of the following minor points.
1. The authors mentioned that amide protons in 1H NMR (Figure 1) were separated into two species by alternative hydrogen bonding with a carbonyl group. However, I think the peak separation is derived from the coexistence of E/Z isomers of the tertiary amide group, and this interpretation is more common.
2. There are some unclear points in the analysis of aggregation in organic solvents. For instance, why do lower concentrations produce larger aggregates? Why are the NMR signals of 3 in DMSO (Figure 2a) sharp and not broad, even though DLS analysis suggests the formation of large aggregates in DMSO? Based on these results, one of the possible explanations is that these compounds exist primarily as monomers in solution, with larger aggregates forming as minor products. Would it be possible to add a discussion on these points to the extent possible?
3. The terms SLN-2 and SLN-3 may not be defined in this manuscript. Please define them if necessary (e.g. SLN-2 = SLN composed of 2).
4. On line 269 or 270, there is a non-English word.
Author Response
First of all, we would like to thank respected Reviewer for careful consideration of the manuscript. In accordance with the comments, the following changes have been made:
- The authors mentioned that amide protons in 1H NMR (Figure 1) were separated into two species by alternative hydrogen bonding with a carbonyl group. However, I think the peak separation is derived from the coexistence of E/Z isomers of the tertiary amide group, and this interpretation is more common.
Thank you for your suggestion. The necessary changes have been made to the manuscript and have been highlighted in green.
Indeed, this separation of proton signals may be due to the coexistence of E/Z isomers, which appear by the tautomerism of the amide group. However, the coexistence of tautomers does not contradict our assumption about the formation of hydrogen bonds by the amide group. The formation of hydrogen bonds was also confirmed by the IR spectra of compound 2 (Fig. S7), in which the narrow intense bands at 2828 and 2935 cm-1 was attributed to the formation of a intramolecular hydrogen bond between amide group and the oxygen atom of the carbonyl fragment.
- There are some unclear points in the analysis of aggregation in organic solvents. For instance, why do lower concentrations produce larger aggregates? Why are the NMR signals of 3 in DMSO (Figure 2a) sharp and not broad, even though DLS analysis suggests the formation of large aggregates in DMSO? Based on these results, one of the possible explanations is that these compounds exist primarily as monomers in solution, with larger aggregates forming as minor products. Would it be possible to add a discussion on these points to the extent possible?
The same behavior was found for macrocycle 2 containing two carboxyl fragments. Critical micelle concentration were established in both solvents (CHCl3 and DMSO) by DLS. The balance of interaction between the dispersed phase and the dispersion medium changes with an increase/decrease in this concentration. In this regard the aggregates become larger or less. It was found that the formation of monodisperse systems does not occur for macrocycle 3 containing one carboxyl function regardless of the concentration and solvent. Apparently, the formation of several types of aggregates and their dynamic equilibrium are observed. The concentrations used in 1H NMR spectra and the concentrations used in DLS to determine the sizes of aggregates are different. The formation of supramolecular polymers is well known for monosubstituted pillar[5]arenes depending on the solvent used [https://doi.org/10.1039/C8CC08085B; https://doi.org/10.1016/j.trechm.2020.07.004]. Some publications show the use of DMSO for preventing the formation of supramolecular polymers [https://doi.org/10.1021/cr500633b; https://doi.org/10.1021/acs.jpcb.5b02515]. It is clearly seen in the 1H NMR spectra, where substituent proton signals resonate as sharp, not broadened signals in the downfield.
- The terms SLN-2 and SLN-3 may not be defined in this manuscript. Please define them if necessary (e.g. SLN-2 = SLN composed of 2).
The necessary definitions of SLN-2 and SLN-3 have been made to the manuscript and have been highlighted in green.
- On line 269 or 270, there is a non-English word.
Thank you! The necessary changes have been made to the manuscript and have been highlighted in green.

Reviewer 3 Report
The manuscript describes the preparation and properties of new derivatives of pillar(n)arenes functionalized with one or two carboxyl groups. This work represents a case example of the broad research field about this class of compounds and molecular aggregates described earlier by J. F. Stoddart et al. (Acc. Chem. Res. 2014, 47, 2631). The novelty of this work is the study of additional two derivates which form the aggregates depending on the substituents on the pillar(n)arene building-block units. In principle, this research could be regarded mostly as a starting point for further investigation of the applicability of these materials. This approach is missing in this work. As such, the manuscript could be interesting mainly for the researches developing syntheses of additional compounds based on this structural unit.
Technical comment: For better characterization of the 1Hnmr, I would recommend to present also 2D nmr spectra.
Author Response
First of all, we would like to thank respected Reviewer for careful consideration of the manuscript. In accordance with the comments, the following changes have been made:
- Technical comment: For better characterization of the 1H NMR, I would recommend to present also 2D NMR spectra.
2D NOESY 1H-1H NMR spectra was recorded for the macrocycle 2. It was added to the manuscript.